# Copy Number Variations Contribute to Intramuscular Fat Content Differences by Affecting the Expression of *PELP1* Alternative Splices in Pigs

**DOI:** 10.3390/ani12111382

**Published:** 2022-05-27

**Authors:** Xia Wei, Ze Shu, Ligang Wang, Tian Zhang, Longchao Zhang, Xinhua Hou, Hua Yan, Lixian Wang

**Affiliations:** 1Key Laboratory of Farm Animal Genetic Resources and Germplasm Innovation of Ministry of Agriculture of China, Institute of Animal Science, Chinese Academy of Agricultural Sciences, Beijing 100193, China; weixia16888@163.com (X.W.); ze_shu@126.com (Z.S.); lffmsfe@126.com (T.Z.); zhlchias@163.com (L.Z.); 7hxh73@163.com (X.H.); zcyyh@126.com (H.Y.); 2State Key Laboratory Breeding Base of Dao-55di Herbs, National Resource Center for Chinese Materia Medica, China Academy of Chinese Medical Sciences, Beijing 100020, China

**Keywords:** intramuscular fat, copy number variations, alternative splicing, *PELP1*, pig

## Abstract

**Simple Summary:**

Copy number variation (CNV) is a type of variant that may influence meat quality of, for example intramuscular fat (IMF). In this study, a genome-wide association study (GWAS) was then performed between CNVs and IMF in a pig F2 resource population. A total of 19 CNVRs were found to be significantly associated with IMF. RNA-seq and qPCR validation results indicated that CNV150, which is located on the 3′UTR end of the proline, as well as glutamate and the leucine rich protein 1 (*PELP1*) gene may affect the expression of *PELP1* alternative splices. We infer that the CNVR may influence IMF content by regulating the alternative splicing of the *PELP1* gene and ultimately affects the structure of the *PELP1* protein. These findings suggest a novel mechanistic approach for meat quality improvement in animals and the potential treatment of insulin resistance in human beings.

**Abstract:**

Intramuscular fat (IMF) is a key meat quality trait. Research on the genetic mechanisms of IMF decomposition is valuable for both pork quality improvement and the treatment of obesity and type 2 diabetes. Copy number variations (CNVs) are a type of variant that may influence meat quality. In this study, a total of 1185 CNV regions (CNVRs) including 393 duplicated CNVRs, 432 deleted CNVRs, and 361 CNVRs with both duplicated and deleted status were identified in a pig F2 resource population using next-generation sequencing data. A genome-wide association study (GWAS) was then performed between CNVs and IMF, and a total of 19 CNVRs were found to be significantly associated with IMF. QTL colocation analysis indicated that 3 of the 19 CNVRs overlapped with known QTLs. RNA-seq and qPCR validation results indicated that CNV150, which is located on the 3′UTR end of the proline, as well as glutamate and the leucine rich protein 1 (*PELP1*) gene may affect the expression of *PELP1* alternative splices. Sequence alignment and Alphafold2 structure prediction results indicated that the two alternative splices of PELP1 have a 23 AA sequence variation and a helix-fold structure variation. This region is located in the region of interaction between PELP1 and other proteins which have been reported to be significantly associated with fat deposition or insulin resistance. We infer that the CNVR may influence IMF content by regulating the alternative splicing of the *PELP1* gene and ultimately affects the structure of the PELP1 protein. In conclusion, we found some CNVRs, especially CNV150, located in *PELP1* that affect IMF. These findings suggest a novel mechanistic approach for meat quality improvement in animals and the potential treatment of insulin resistance in human beings.

## 1. Introduction

Intramuscular fat (IMF) refers to the total amount of fat located between muscle fibers and as lipid droplets in the muscle cells. In humans, extra IMF deposition has been reported to be associated with type 2 diabetes and insulin resistance [1]. In animals, IMF content directly influences the flavor and juiciness of meat and indirectly influences the tenderness and color of meat [2]. Because pork IMF contains many long-chain polyunsaturated fatty acids, it can also directly affect human health [3].

IMF content is a quantitative trait with low-to-moderate heritability (0.2–0.4) [4,5] and is influenced by multiple genes or quantitative trait loci (QTLs) in animals. As stated in the PigQTLdb (http://www.animalgenome.org/cgi-bin/QTLdb/SS/index, accessed on 23 August 2021) a total of 842 IMF-related QTLs have been reported [6]. Although thousands of single-nucleotide polymorphisms (SNPs) have been reported to be associated with IMF in genome-wide association studies (GWAS), most of them only explain a small part of the total genetic variance [4,7,8].

Copy number variations (CNVs) are the mutation type with the widest coverage of the genome and are one of the variation types most likely to explain “missing inheritance” beyond the SNP effect. In pigs, there have been several studies revealing CNVs or CNV regions (CNVRs) associated with economic traits. In the research of Rubin et al., CNVs of *KIT* were reported to be associated with coat color [9]. Fowler et al. (2013) found a CNV region related to backfat thickness on chromosome 7 using a pig 60K Beadchip [10]. In the research of Revilla et al. (2017), Chen et al. (2018), Stafuzza et al. (2019), and Krüger et al. (2020), CNVs were reported to be associated with fatty acid composition and growth traits, ear size, number of piglets born alive, and prevalence of porcine endogenous retroviruses [11,12,13,14]. In our previous research, CNVRs have reported associations with IMF content [15]. The previous research used 60K Beadchip data with only 48 CNVs detected.

In this study, we used genome-wide CNV markers identified and genotyped via next-generation sequencing to perform genome-wide association analysis on IMF. Moreover, function analyses of these CNVs were also performed to analyze the potential effects and mechanisms of the CNVs on IMF. We aimed to identify the candidate genes and causal mutation sites of IMF and provide a research basis for quality pig meat breeding.

## 2. Materials and Methods

### 2.1. Animals and Phenotype Determination

The pigs used in the experiment were all from the Large white pig × Min F2 resource population, raised in the Changping pig farm of the Institute of Animal Science, Chinese Academy of Agricultural Sciences. All 592 F2 individuals were raised to market age (240 ± 7 days) and slaughtered for commercial purposes. The F2 individuals were gilts or barrows and have good health status. All of the F0 individuals were raised as normal sows, and ear tissues were collected for DNA extraction when they were one-day old. *Longissimus dorsi muscles* (LDM) were collect for IMF content measurement and DNA sequencing for F2 individuals. IMF content was measured using an ether extraction method (Soxtec Avanti 2055 Fat Extraction System; Foss Tecator, Hillerød, Denmark).

### 2.2. Tissue DNA Extraction and Sequencing

DNA of LDM tissue (for F2 individuals, 592 individuals) and ear tissue (for F0 individuals, 19 individuals) was extracted using the phenol imitation method. The NANODROP 1000 was used to detect the concentration and quality of DNA (Thermo Scientific, Waltham, MA, USA). An Illumina Hi-seq2500 was used for paired-end sequencing, and the sequencing depth was 5–7× for F2 individuals, 30× for F0 individuals.

### 2.3. CNV Detection, Genotyping, and Genomic Association Analysis

CNVcaller software was used to detect and genotype all individual CNVs. First, the reference genome (version Sus scrofa 11.1, Ensembl, USA) is segmented into overlapping sliding windows of 500 bp [16]. Second, the reads of each window were counted across the genome from the BAM file and generate a comparable read depth (RD) file of each individual. In the third step, we used a strict standard of silhouette_score > 0.6 for CNV determination. At last, when genotyping, each CNV region was divided into no more than three genotypes, and each sample was labeled 1 (gain), 2 (normal), and 3 (loss) to represent the genotype cluster to which the individual belonged.

CNV-based GWAS was performed using the GLM, mixed linear model (MLM), and FarmCPU model approaches with rMVP software [17]. In the GLM method, we used the covariate of sex and batches. The model can be written as follows:*y* = *Xb* + *Zu* + *e*
where *y* is the vector of phenotypic values; *b* is a vector of fixed effects; and *u* represents breeding values. *X* and *Z* were design matrices for *b* and *u*, respectively; *e* represents the residual error vector.

In the FarmCPU method, we used principle component analysis (PCA) as a fixed effect. The model can be written as follows:*y* = *Tw**_i_* + *P**_j_**q**_j_* + *m**_k_**h**_k_* + *e*
where *y* is the vector of phenotypic values; *T* is a matrix of fixed effects, *w_i_*; *P_j_* is the genotype matrix of *j* pseudo quantitative trait nucleotides (QTNs), which was used as the fixed effect; and *q_j_* is the corresponding effect. *m_k_* is a vector of genotypes for the kth marker to be tested, and *h_k_* is the corresponding effect. *e* is the residual effect vector.

In the method of MLM, we also used PCA as a fixed effect, and the MLM model can be written as follows:*y* = *Wb* + *Za* + *Sc* + *e*
where *y* is the vector of phenotypes of each pig, *a* is the vector of fixed effects, *b* is the vector of the SNP substitution effects, and *c* is the vector of random additive genetic effects with *a~*N(0, G*_σ2a_*). *W*, *Z*, and *S* are the incidence matrices for *b*, *a*, and *c*, respectively.

The selected threshold for genomic level significance was −log10(0.05/N), where N was the total number of CNVRs after quality control.

### 2.4. RNA Data Acquisition

Six RNA-seq datasets of *longissimus dorsi muscle* tissue were used for RNA expression analysis, as in our previous research [18]. These data have been submitted to the Genome Sequence Archive with the accession number CRA001645.

### 2.5. Validation of the CNVs and PELP1 RNA Sequencing Results Using qPCR

DNA of five individuals in each CNVR genotype group were used for qPCR amplification for CNVR validation. The primers were designed using the Primer 6 software. The glucagon gene (*GCG*) was used as the single-copy control. RNA extract from five individuals’ LDMs in each CNVR genotype group was used for expression validation. The glyceraldehyde-3-phosphate dehydrogenase (*GAPDH*) gene was used as the housekeeping gene. Copy number and relative expression were calculated using the 2^−ΔΔCT^ method [19], where ΔCT = CT(target region cycle) − CT(control region cycle), ΔΔCT = ΔCT(target) − ΔCT(standard control). Moreover, 2^−ΔΔCT^ stood for the comparison of the ΔCT value of the samples with CNV to those without CNV. The PCR cycle was as follows: 3 min at 50 °C, 10 min at 95 °C, 40 cycles of 15 s at 95 °C, and 1 min at 60 °C. The primer sequences are shown in Appendix A.

In the CNV’s validation analysis, four CNVRs which were CNV11, CNV148, CNV150, and CNV223, were compared independently between the normal and Duplicate/Deleted group using a T-test in SAS software (version 9.2). In the *q*PCR amplification analysis, the expression of the two *PELP1* alternative splices (ENSSSCT00000075280 and ENSSSCT00000019507) was compared using a *t*-test in SAS software (version 9.2). *p* < 0.05 was considered to indicate a significant difference.

### 2.6. Annotation of CNVRs

For the combined analysis of CNVRs and QTLs, QTLs were retrieved from the PigQTLdb [6]. Bedtools (v2.27.1) [20] software was used, and the “intersection” command was used: intersectBed -a -b -wa -wb > QTLs.txt. The PPI network contained the top 25 interactions of PELP1 retrieved from the STRING website: (https://version11.string-db.org/cgi/network.pl?taskId=NzfSlslTYWvr, accessed on 12 August 2021) [21].

### 2.7. Protein Alignment and Structure Prediction

Consensus sequences and alignment of A0A5G2R420 and F1RFT3 were built using ClustalX in MEGA X [22]; the structures of these two sequences were predicted using alphafold2 [23] and visualized using iCn3D [24].

## 3. Results

### 3.1. Phenotypic Distribution of Pig IMF Content

For the 592 F2-generation individuals of the Large white pigs × Min pig resource population, the average slaughter weight (240 ± 7 d) was 109.13 kg, and the average carcass weight (left side) was 40.38 kg. The average IMF content of these pigs was 2.87%, the maximum was 12.70%, the minimum was 0.73%, the standard deviation was 1.85%, and the coefficient of variation was 0.65.

### 3.2. Porcine Genome Copy Number Variation Segmentation

CNVRs were separated by calling and genotype in the F0 and F2 population, and all the CNVRs were screened with the silhouette_score > 0.6 in this study. CNVRs could be investigated in both the F0 and F2 population we selected for further research. Finally, a total of 1185 CNV regions (CNVRs) were identified, including 393 duplicated CNVRs, 432 deleted CNVRs, and 361 CNVRs with both duplicated and deleted status (Figure 1). The length of CNVRs ranged from 2000 bp to 4,196,500 bp, with an average length of 30,818 bp. The numbers of copies were relatively evenly distributed on autosomes and basically corresponded to the lengths of the chromosomes.

### 3.3. CNV Association Analysis

All 592 pigs were used in a genome-wide association study (GWAS) between CNVs and IMF content. The GWAS results are shown in Table 1, and the Manhattan plot of the GWAS analysis is shown in Figure 2. Only 1 CNVR significantly associated with IMF could be identified using the general linear model (GLM) method, and 19 CNVRs significantly associated with IMF were identified using the fixed and random model circulating probability unification (FarmCPU) method. Phenotype comparison among the different CNV genotypes of 19 significant CNVs were shown in Appendix A. The significant CNVR detected by the overlapped two methods was located on chromosome 12: 52,194,501–52,220,000. The other CNVRs were located on chromosome 1 (2 CNVRs), chromosome 2 (3 CNVRs), chromosome 3 (1 CNVR), chromosome 4 (1 CNVR), chromosome 7 (2 CNVRs), chromosome 8 (1 CNVR), chromosome 12 (3 CNVRs), chromosome 13 (1 CNVR), chromosome 14 (1 CNVR), chromosome X (2 CNVRs), and chromosome Y (1 CNVR). Among the 19 total CNVRs, 8 CNVRs were deletion CNVRs, and the others were duplication CNVRs.

### 3.4. Quality Assessment of CNVs Using Quantitative Real-Time PCR (qPCR)

In this study, DNA qPCR validation was performed to confirm the existence of the identified CNVRs. Data of each group could meet the assumptions of homogeneity of variance (*p* > 0.05) and normal distribution (*p* > 0.05). As shown in Figure 3A–D, all of the selected four CNVRs were validated by DNA qPCR.

### 3.5. Annotation and QTLs’ Co-Location with CNVRs

In order to study the potential function of the CNVRs, we first annotated the CNV coverage area. As shown in Table 1, there were nine known genes (including four olfactory receptor genes) overlapped with six CNVRs and four novel genes overlapped with four CNVRs. A total of 10 of the 19 CNVRs were located in the intergenic region, without covering any genes. Considering the CNVRs and 842 known IMF-associated QTLs published in all previous studies together, 3 of the 19 significant CNVs could be overlapped with at least one of the known QTLs for IMF (Appendix A). However, these 3 CNVs were not overlapped with IMF-associated QTLs or SNPs investigated in our previous using the same population (Appendix A).

### 3.6. Significantly Related CNV Internal RNA-Seq

Six individuals with different copy numbers were selected for RNA sequencing to predict the potential effect mechanisms of the CNVRs. The relationships between DNA dosage and RNA expression were then analyzed. The results of the transcript expression levels for the 13 overlapped genes showed that related transcript expression differences could be detected only in the Proline, Glutamate, and Leucine Rich Protein 1 (*PELP1*) genes. The expression of one of the PELP1 alternative splices (ASs), ENSSSCT00000019597, was significantly different between different individual CNVRs (Table 2).

### 3.7. Functional Analysis of CNV150

As our research population was constructed using Large white pigs and Min pig F0 individuals, which have different IMF contents (the average IMF content of the Large white is less than 2%, and the average IMF content of Min pigs is more than 4%), we also detected the copy numbers of F0 individuals to confirm the relationship between IMF and CNVR150. From Figure 4, we can see that in Large white pigs, the copy number of CNV150 was multiple, and in Min pigs, the copy number of CNV150 was around 1.

Beyond the RNA-seq data, we also detected the expression of the two PELP1 alternative splices (ENSSSCT00000075280 and ENSSSCT00000019507) using the qPCR method. As shown in Figure 3E,F, the expression of ENSSSCT00000019507 was significantly different between normal-copy-number individuals and duplicated-copy-number individuals. ENSSSCT00000075280 and ENSSSCT00000019507 were coding modulators of the nongenomic activity of the estrogen receptor proteins A0A5G2R420 and F1RFT3.

In order to mine the mechanism of CNV150 in depth, we analyzed the interaction network of PELP1. From the protein–protein interaction (PPI) networks (Figure 5), a total of 23 proteins were experimentally determined to interact with PELP1 (Appendix A). Among the 23 proteins, the androgen receptor (AR), estrogen receptor (ESR1), glucocorticoid receptor (GR, NR3C1), nuclear receptor subfamily 4 group A member 1(NR4A1), retinoblastoma-associated protein (RB1), 60S ribosomal protein L11 (RPL11), proto-oncogene tyrosine-protein kinase Src (SRC), and signal transducer and activator of transcription 3 (STAT3) have been reported to be related to fat deposition, metabolism, or insulin resistance.

In order to study whether the protein structure variation affected the interaction between PELP1 and its interactive proteins, we aligned the AA of A0A5G2R420 and F1RFT3 and found there was a 23 AA (83–105) difference between these two proteins (Figure 6A). Alpha fold 2 was used to predict the potential structure of these two sequences. In F1RFT3 (coded by ENSSSCT00000019597), a helix (about 30 AA) is unfolded as compared to A0A5G2R420 (coded by ENSSSCT00000075280) (Figure 6B,C). In A0A5G2R420, the structure located in the variation region had a very high confidence (predicted LDDT (pLDDT) > 90). This helix is located between two LXXLL motifs (located on 69–74 AA and 111–116 AA) of PELP1.

## 4. Discussion

The Min is a native Chinese pig breed with an average IMF of > 4.0%, and the Large white is a commercial pig breed with an average IMF of ≤2.0%. The Large white × Min F2 separated population is an ideal population in which to investigate the candidate genes or QTLs for IMF. There is a huge variation in the IMF in our population (0.73%~12.7%). As we known, the “segregation variance” can often be observed in the F2 generation, and the magnitude of segregation variance depends on the extent of population differentiation and the genetic base [25,26]. In this study, we first used NGS data of the F2 resource population for CNV calling and genotyping, and then performed CNV-based GWAS for candidate CNV identification. CNV calling using NGS data mainly uses four approaches: paired-end mapping, split-reads, sequence assembly, and RD [27]. CNVcaller software [16], which uses the RD method, could mitigate the influence of high proportions of gaps and misassembled duplications in the nonhuman reference genome assembly for CNV calling and genotyping and was suitable for our population. In our research, a total of 1185 CNVRs were identified, and this number was smaller than some other pig CNVRs detection research. For example, in the research of Zheng et al., a total of 12,668 CNVRs were detected in 32 Meishan pigs and 29 Duroc pigs [28], and in the research of Wu et al., a total of 18,687 CNVs were identified in Tongcheng and Large white pigs [29]. This may have been caused by the strict standard for the CNV definition (silhouette coefficient > 0.6) we used. The qPCR validation results indicated that the selected CNVRs were all real.

In our study, only FarmCPU detected serval significant signals; the other two methods detected no signal or only one significant signal. FarmCPU could split MLM into a fixed effect model (FEM) and a random effect model (REM), using them iteratively to increase the power for detecting candidate genes associated with the population structure. Association tests in FarmCPU are validated by FEM with the same computing efficiency as GLM, while the statistical power surpasses that of MLM at the same level of type I errors. In much research, FarmCPU could detect more significant signals than MLM and GLM could in our research [4,17,30].

QTL co-location is a method to analyze the potential functions of unknown variations or regulation transcriptional factors [31]. In order to study the function of the CNVRs located in the intergenic region or in novel genes, we also performed QTL co-location analysis in this study, and ultimately found that 3 of the 19 CNVRs were located in the regions of reported IMF-associated QTLs. We infer that these QTLs may affect IMF through structure variation or just a marker linked to the IMF traits. Further experimental research was also required.

In order to identify the potential function of CNVRs, the function of the known genes in which significant CNVRs were located was retrieved. Among the nine genes, *PELP1* is an ESR coregulator protein [32] and has been reported to be associated with sperm morphology abnormalities in pigs [33]. In the human and great ape *PELP1* gene, duplicated CNVs also exist [23,34]. Homozygous spermatogenesis associated 22 (*SPATA22*) is a sex-related gene associated with infertility and related traits [35]. The SH3 domain-binding protein 4 (SH3BP4) is a negative regulator of amino acid-Rag GTPase-mTORC1 signaling and is related to diabetic retinopathy [36,37]. The FCH And Mu Domain Containing Endocytic Adaptor 2 (FCHO2) protein can participate in the early step of clathrin-mediated endocytosis and has lipid-binding activity [38]. Other known genes, namely LOC100524322, LOC100524156, and R-SSC-381753, are olfactory receptors (ORs), and ORs have been reported to be related to IMF or insulin resistance in previous research [39,40,41,42].

As CNVRs usually work through regulation effects or dose effects [43,44], we analyzed the RNA expression profiles of some individuals with different CNVR dosages. Interestingly, we found that one of the *PELP1* ASs, named ENSSSCT00000019597, was significantly differently expressed in CNV150-variant individuals. We then validated the differential expression using *q*PCR and the results were positive. Hence, we inferred that this CNV150 may affect *PELP1* alternative splicing.

In order to confirm the function of CNV150, we then analyzed the read depth of CNV 150 in F0-generation individuals and found that the copy number of CNV150 was normal in Min pigs and duplicate in Large white pigs. As shown in Table 1, this CNVR has a negative effect for IMF, and Min pigs and Large white pigs are high- and low-IMF pigs, respectively. These results were consistent. Moreover, there is a QTL which is associated with meat color a*, percentage type IIb fibers, adipocyte diameter, and marbling that could overlap with CNV150 [45,46,47]. So, we then further studied *PELP1* and its ASs.

First, we studied whether *PELP1* directly or indirectly affects IMF. In the PPI networks, about half of the proteins had been reported as being related to IMF or insulin resistance. Among these genes or proteins, AR and ESR1 can regulate leptin transcript accumulation and protein secretion in adipocytes [48]. The NR3C1 transcription factor has been identified as a potential regulator co-localizing within QTLs for fatness and growth traits [49]. NR4A1 can affect insulin resistance and downregulated intramuscular lipid content [50]. RB1 has a direct interaction relationship related to adipogenesis growth [51]. RPL11 has been revealed to play a role in fat storage [52]. SRC and STAT3 can respond to adipogenesis through the TNF-α pathway [53]. Thus, we inferred that PELP1 may influence IMF by interacting with other proteins.

In previous research, the interacting regions of PELP1 and ESR, AR, GR, RB, and STAT3 were amino acids 1–400, LXXLL motifs, amino acids 1–600, or amino acids 1–330 [54]. Hence, we then studied whether the ASs affected the 3D structure of the PELP1 protein and affected the interaction between PELP1 and its interactive proteins. The results predicted by Alphafold2 indicated that, in the variation location of amino acids 83–105, a helix was unfolded in F1RFT3. Moreover, this helix was between two LXXLL motifs. Alphafold2 has been used to predict the structures of many difficult protein targets at or near experimental resolution, and the results have high reliability [23]. Thus, we inferred that the structure changes potentially caused by CNV150 may affect the interaction of PELP1 and its interactive proteins. However, the function and molecular regulatory mechanisms between CNV150 and IMF content require further experimental research, such as gene knockdown/editing, co-immunoprecipitation, and so on.

## 5. Conclusions

In this study, a CNV-based GWAS was performed between CNVs and IMF, and a total of 19 CNVRs were found to be significantly associated with IMF. Some CNVRs, such as the three CNVRs overlapped with known QTLs, may be useful candidate markers for IMF selection. CNV150, which was located on the 3′UTR of PELP1, may influence IMF content by regulating the alternative splicing of the PELP1 gene and the structure of the PELP1 protein. These findings suggest a novel mechanistic approach for meat quality improvement in animals and the potential treatment of insulin resistance in human beings.

## Figures and Tables

**Figure 1 animals-12-01382-f001:**
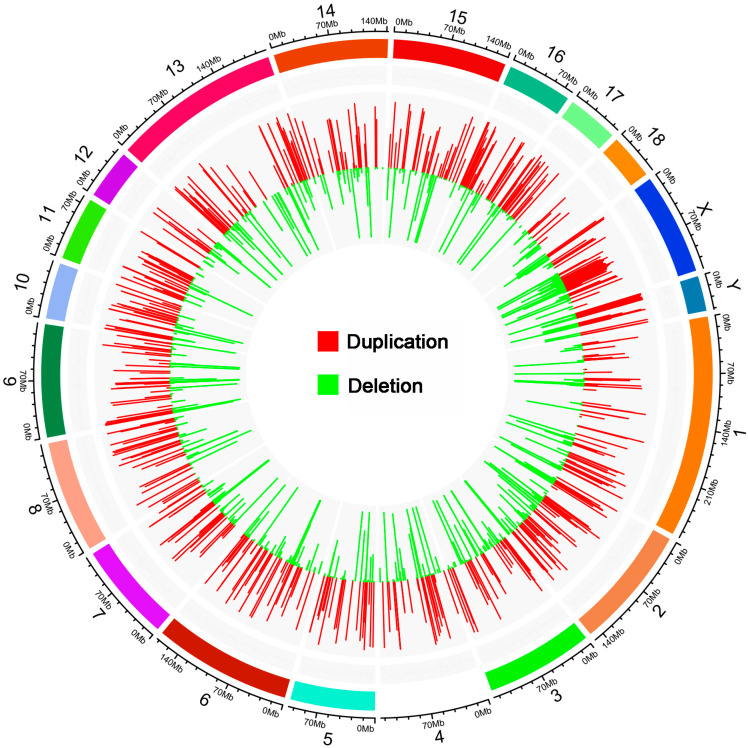
The distribution of pig CNVRs. The outer circle presents the lengths of the chromosomes; the inner circle presents the CNVR distributions. The length of the histogram bars indicates the ratio of different types of CNVRs to the total number of CNVRs.

**Figure 2 animals-12-01382-f002:**
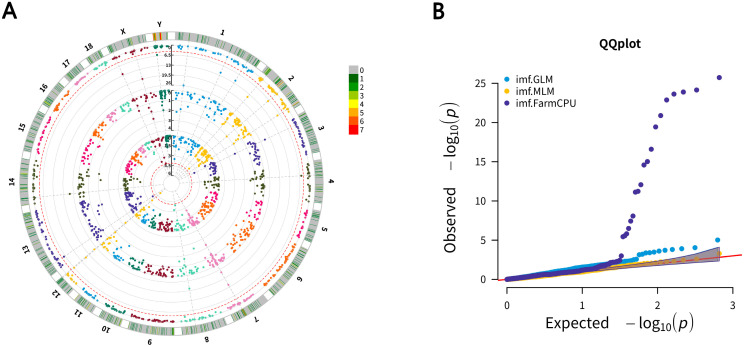
Circular Manhattan plot and quantile–quantile (QQ) plot of associated CNVs for IMF. (**A**) Circular Manhattan plot of associated CNVs for IMF using GLM, MLM, and FarmCPU methods; the outer circle is the number of CNVRs in the 1 Mb size region. Significant CNVRs in the same location are marked in lines; (**B**) QQ plot of CNVs associated with IMF using GLM, MLM, and Farm-CPU methods.

**Figure 3 animals-12-01382-f003:**
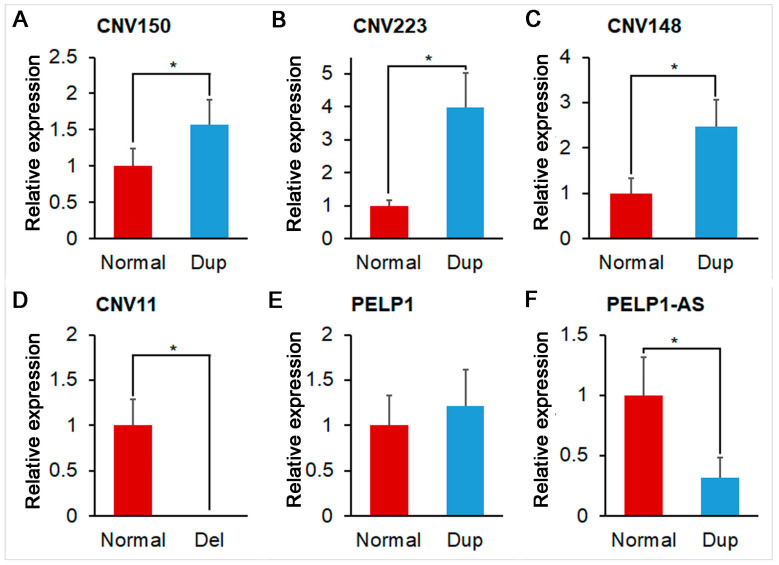
Relative expression of validated CNVRs and alternative splices of PELP1. (**A**) Relative expression of normal and duplicate (Dup) status of CNV150; (**B**) Relative expression of normal and duplicate (Dup) status of CNV223; (**C**) Relative expression of normal and duplicate (Dup) status of CNV148; (**D**) Relative expression of normal and deletion (Del) status of CNV11; (**E**) Relative expression of normal and duplicate (Dup) status of PELP1 alternative splice 1 (ENSSSCT00000075280); (**F**) Relative expression of normal and duplicate (Dup) status of PELP1 alternative splice 2 (ENSSSCT00000019507). * represents significant difference (*p* < 0.05).

**Figure 4 animals-12-01382-f004:**
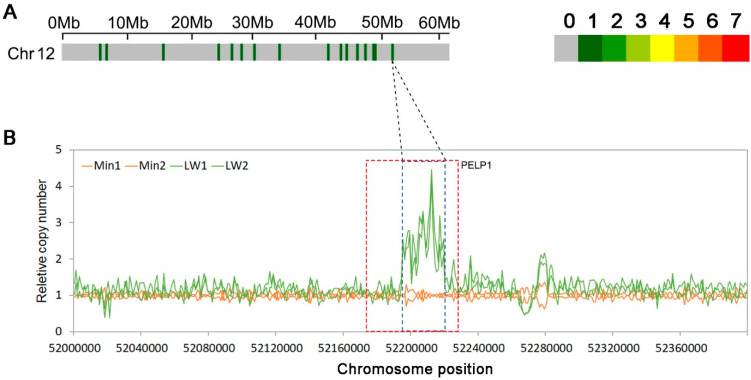
CNVRs on chromosome 12 in Large white and Min pigs. (**A**) CNVRs on pig chromosome 12, where the colored lines represent the numbers of CNVRs in the 1 Mb region; (**B**) Relative copy numbers in the 1-kb sliding windows across the CNV150 peak region for Large white (Green) and Min pigs (Orange). The region in the red box represents the position of gene *PELP1* (Proline, Glutamate, and Leucine Rich Protein 1).

**Figure 5 animals-12-01382-f005:**
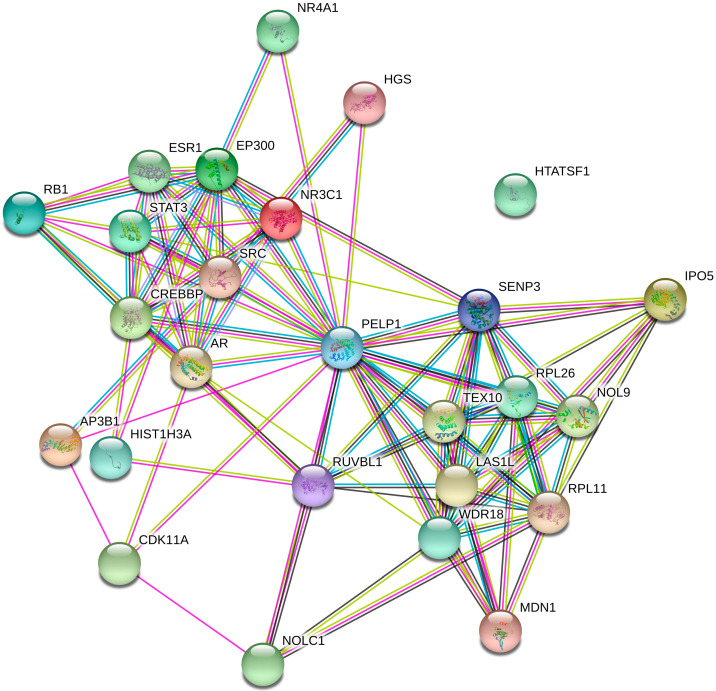
PPI network of PELP1. This figure displays data obtained from three databases and illustrated using STRING.

**Figure 6 animals-12-01382-f006:**
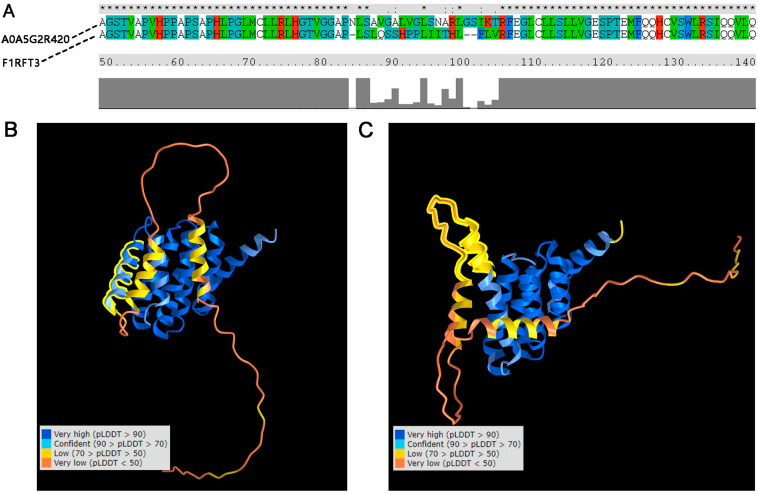
Comparison of the sequence and structure of PELP1 alternative-splice proteins. (**A**) Sequence alignment of the two proteins. A0A5G2R420: protein translated by ENSSSCT00000075280. F1RFT3: protein translated by ENSSSCT00000019507; (**B**) 3D structure of A0A5G2R420, where the structure marked by a yellow line indicates a helix with very high confidence (predicted LDDT (pLDDT) > 90); (**C**) 3D structure of F1RFT3, where the structure marked by a yellow line indicates an unfolded helix with low confidence (70 > pLDDT > 50).

**Table 1 animals-12-01382-t001:** Descriptions of the significant CNVRs associated with IMF.

CNVRs	Chromosome	Start	End	Effect	*p* Value	Type	Overlapped Gene
CNV150	12	52,194,501	52,220,000	−0.4987	9.47 × 10^−6^	Dup	*PELP1*
CNV11	1	43,146,501	43,151,500	−0.5285	1.85 × 10^−26^	Del	-
CNV150	12	52,194,501	52,220,000	−0.4553	7.08 × 10^−25^	Dup	*PELP1*
CNV657	7	79,216,001	79,273,500	0.6623	1.29 × 10^−24^	Dup	ENSSSCG00000035754
CNV223	14	2,0385,001	20,388,500	0.7767	2.35 × 10^−24^	Dup	-
CNV466	3	25,786,001	25,790,000	0.6017	1.32 × 10^−23^	Dup	-
CNV698	8	91,666,001	91,675,500	0.3071	1.30 × 10^−21^	Del	-
CNV846	X	75,184,001	75,198,000	0.3922	3.61 × 10^−20^	Dup	ENSSSCG00000046526
CNV149	12	49,461,001	49,498,500	−0.2625	2.53 × 10^−17^	Dup	SPATA22
CNV385	2	17,955,001	17,967,500	0.3497	9.45 × 10^−16^	Dup	-
CNV653	7	78,876,501	78,952,500	−0.3832	2.58 × 10^−15^	Dup	SH3BP4, LOC100524322, LOC100524156, ENSSSCG00000044162, R-SSC-381753
CNV771	X	8,710,001	88,350,00	0.4339	8.49 × 10^−13^	Dup	-
CNV49	1	236,168,001	236,172,000	−0.2828	5.89 × 10^−12^	Del	ENSSSCG00000049310
CNV35	1	174,451,501	174,455,500	−0.3230	7.50 × 10^−12^	Del	-
CNV450	2	142,722,001	142,725,000	0.2647	8.33 × 10^−9^	Del	-
CNV422	2	82,146,001	82,150,000	−0.3249	3.70 × 10^−8^	Dup	FCHO2
CNV148	12	49,181,501	49,205,000	0.2101	3.19 × 10^−7^	Del	ENSSSCG0000034084
CNV901	Y	5,446,501	5,449,000	−0.2768	1.54 × 10^−6^	Dup	-
CNV160	13	25,532,501	25,537,500	−0.1775	2.77 × 10^−6^	Del	ULK4
CNV508	4	44,804,501	44,809,500	−0.2026	3.61 × 10^−6^	Dup	-

Dup: duplicate; Del: deletion; *PELP1*: Proline, Glutamate, and Leucine Rich Protein 1; *SPATA22:* Spermatogenesis Associated 22; *SH3BP4:* SH3 Domain Binding Protein 4; *FCHO2*: FCH And Mu Domain Containing Endocytic Adaptor 2; and *ULK4*: Unc-51 Like Kinase 4.

**Table 2 animals-12-01382-t002:** Reads counts of the PELP1 alternative splices.

Individual	ENSSSCT00000075280 Reads Count	ENSSSCT00000019507 Reads Count	Copy Numbers
H1	2090	12	Normal
H2	1986	240	Normal
H3	2121	58	Normal
L1	2501	858	Duplicated
L2	2298	436	Duplicated
L3	1984	867	Duplicated

## Data Availability

The datasets presented in this study can be found in online repositories. The sequencing data used in the current study have been submitted to the Genome Sequence Archive, with the accession number CRA002451.

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
