# Peer review of "Copy Number Variations Contribute to Intramuscular Fat Content Differences by Affecting the Expression of PELP1 Alternative Splices in Pigs"

_animals, 2022, doi:10.3390/ani12111382_

Round 1
Reviewer 1 Report
The manuscript reports an association study between CNVs and IMF in an F2 population.
The manuscript is interesting even if it should be improved in several parts.
1) M&M: the number of F0 and F2 sequenced animals should be reported.
2) M&M: the method to call CNV should be better described - please refer to gains and losses
3) M&M: the methods used in the GWAS should be better described -
4) M&M: The tissue(s) used for RNAseq and expression analysis are not indicated (muscle? which muscle?)
5) M&M: please better describe the expression gene analysis
6) Results - it is not clear if QTL identified with CNVs identified in this study map in the same regions of the QTL odentified in the previous studies of the same group which were based on the same population - both using CNV and SNP based GWAS. They might overlap as linkage disequilibrium in this population is high. The association between CNVs and IMF in this study does not ean that CNV might be involved in affecting this trait. This part should be clearly considered and the results and discussion modified accordingly
Author Response
The manuscript reports an association study between CNVs and IMF in an F2 population.
The manuscript is interesting even if it should be improved in several parts.
1) M&M: the number of F0 and F2 sequenced animals should be reported.
Response: Follow your suggestion, we have added the number of F0 and F2 sequenced animals (Lines 85-86).
2) M&M: the method to call CNV should be better described - please refer to gains and losses
Response: Follow your suggestion, we have added the calling procedure and the reference parameter used in each procedure of CNVcaller. (Lines 91-98).
3) M&M: the methods used in the GWAS should be better described -
Response: Follow your suggestion, we have added the description of models which used in GLM, MLM, and FarmCPU method for GWAS in this paper. (Lines 101-118).
4) M&M: The tissue(s) used for RNAseq and expression analysis are not indicated (muscle? which muscle?)
Response: For RNAseq and expression analysis we used the tissue of longissimus dorsi muscle. Follow your suggestion, we have added this in the RNAseq analysis (Line 123). And there already a description for expression analysis (Line 130).
5) M&M: please better describe the expression gene analysis
Response: In the CNVs validation analysis, four CNVRs which were CNV11, CNV148, CNV150, and CNV223 were compared independently between the normal and Duplicate/Deleted group using T-test in SAS software (version 9.2). In the qPCR amplification analysis, the expression of the two PELP1 alternative splices (ENSSSCT00000075280 and ENSSSCT00000019507) was also compared using T-test in SAS software (version 9.2). P < 0.05 was considered to indicate a significant difference. Follow your suggestion, we have added this description in current version of manuscript. (Lines 138-144).
6) Results - it is not clear if QTL identified with CNVs identified in this study map in the same regions of the QTL odentified in the previous studies of the same group which were based on the same population - both using CNV and SNP based GWAS. They might overlap as linkage disequilibrium in this population is high. The association between CNVs and IMF in this study does not ean that CNV might be involved in affecting this trait. This part should be clearly considered and the results and discussion modified accordingly
Response: In this study, we did not use the same populations in previous studies to do overlapped study. Instead, we download the whole IMF associated QTLs published in all previous studies, and then mapped with our significant CNVRs. As you mentioned, we also think the association between CNVs and IMF in this study does not mean that CNV might be involved in affecting this trait. We only using this analysis to annotate the potential function of CNVRs. Follow your suggestion, we have clearly described this in the results (Lines 215-219) and modified the description in the discussion part. (Lines 311-317).
Reviewer 2 Report
Dear Authors
I received a thesis for review entitled: Copy Number Variations Contribute to Intramuscular Fat Content Differences by Affecting the Expression of PELP1 Alternative Splices in Pig
Overall, the research direction is valid and the results are interesting, but the publication requires some improvement.
L: 14 There is an extra space at the end of the sentence
L: 81 The Latin name of the muscles should be written in italics
L: 119 No statistic program name information available. Only the tests used are described. It is also difficult to state the legitimacy of their choice, as there is no information on the data. Did the results meet the assumptions of homogeneity of variance and normal distribution?
L: 135 What could be the reason for such a huge variation in the IMF results, ie from 0.73 to 12.70 %? It would be worth mentioning this in the discussion of the results.
Discussion is the weakest point of the publication. Most of the information is a methodology or description often deviating from the mainstream of research, more suitable for the "Introduction" chapter. I believe that this section should be thoroughly redrafted and focus more on the interpretation of the results and their comparisons with other studies.
L: 254-255 It's more like "Methods" not "Discussion". The same information is already in L: 91
L: 265-272 This is the information that should be in the "Results" section
L: 268 There is "Farmpcu", but it should be "FarmPCU" according to the accepted nomenclature
L: 284 There is an extra space before the word "Other"
L: 278-286 This is rather content matching the "Introduction" chapter
There are many examples of this. After thorough improvements, I believe the publication is worth considering for publication in the Animals.
Author Response
I received a thesis for review entitled: Copy Number Variations Contribute to Intramuscular Fat Content Differences by Affecting the Expression of PELP1 Alternative Splices in Pig
Overall, the research direction is valid and the results are interesting, but the publication requires some improvement.
L: 14 There is an extra space at the end of the sentence
Response: Done. (Line 14)
L: 81 The Latin name of the muscles should be written in italics
Response: Done. (Line 81)
L: 119 No statistic program name information available. Only the tests used are described. It is also difficult to state the legitimacy of their choice, as there is no information on the data. Did the results meet the assumptions of homogeneity of variance and normal distribution?
Response: After double check, we confirmed that in the CNVs validation analysis, four CNVRs which were CNV11, CNV148, CNV150, and CNV223 were compared independently between the normal and Duplicate/ Deleted group using T-test in SAS software (version 9.2). In the qPCR amplification analysis, the expression of the two PELP1 alternative splices (ENSSSCT00000075280 and ENSSSCT00000019507) was also compared using T-test in SAS software (version 9.2). P < 0.05 was considered to indicate a significant difference. Data of each group could meet the assumptions of homogeneity of variance (P>0.05) and normal distribution (P>0.05). Follow your suggestion, we have added this description in current version of manuscript. (Lines 138-144 and Lines 198-199).
L: 135 What could be the reason for such a huge variation in the IMF results, ie from 0.73 to 12.70 %? It would be worth mentioning this in the discussion of the results.
Response: The individuals used in our study were from a F2 population which was constructed using two breeds of pigs with extremely different average IMF. As we known, the “segregation variance” can often be observed in the F2 generation. The magnitude of segregation variance depends on the extent of population differentiation and the genetic base (Lande et al. 1981, Slatkin and Lande 1994). That may be the reason why a huge variation in the IMF appeared in our population. Follow your suggestion, we have added this disscussion and the reference which support our view in the manuscript. (Line 283-286)
Lande R. The minimum number of genes contributing to quantitative variation between and within populations. Genetics. 1981
Slatkin M, Lande R. Segregation variance after hybridization of isolated populations. Genet Res. 1994 Aug;64(1):51-6.
Discussion is the weakest point of the publication. Most of the information is a methodology or description often deviating from the mainstream of research, more suitable for the "Introduction" chapter. I believe that this section should be thoroughly redrafted and focus more on the interpretation of the results and their comparisons with other studies.
Response: Follow your suggestion, we have made a major modification of the discussion section. First, we have moved some description you mentioned to the method or results section and deleted some duplicated description (details were as follow responses). Second, we have adjusted the order of some paragraphs to make the logic more coherent (Lines 311-317). Third, we have added some discussion about our purpose of analysis (Lines 311-312, Lines 318-319, Lines 368-370). Fourth, other descriptions which more suitable for results was also moved or adjusted (Lines 266-271, Lines 360-370). Fifth, we have added some references to support our study (reference 25,26,32, 44, and 45).
L: 254-255 It's more like "Methods" not "Discussion". The same information is already in L: 91
Response: Follow your suggestion, we have deleted this sentence in the Discussion section and made corresponding modification of the following sentence (Lines 289-291)
L: 265-272 This is the information that should be in the "Results" section
Response: Follow your suggestion, we have removed this description to the Results section (Line 181)
L: 268 There is "Farmpcu", but it should be "FarmPCU" according to the accepted nomenclature
Response: Farmcpu has been revised to FarmCPU. (Line 303 and Line 309)
L: 284 There is an extra space before the word "Other"
Response: Done (Line 327)
L: 278-286 This is rather content matching the "Introduction" chapter
Response: In this paragraph we mainly discuss the potential function of CNVRs by analysis the CNVRs located genes. And another purpose of this paragraph is to introduce the next paragraph. So, we think it is not suitable to move to “Introduction” chapter. Instead, we have added a head to describe the purpose of this analysis. (Lines 318-320)
Round 2
Reviewer 1 Report
The manuscript has been improved. Figure 4 should indicate the position of the pelp1 gene.
Author Response
Follow your suggestion, we have added the position of the PELP1 gene in Figure 4